# Dataset Distillation for Domain Generalization

## Abstract

Dataset Distillation (DD) has been applied to various downstream tasks and recently scaled to ImageNet-1k, highlighting its potential for practical applications. However, in real-world scenarios, robustness to unseen domains is essential, and the robustness of models trained on synthetic datasets remains uncertain. To address this, we propose a novel task, Dataset Distillation for Domain Generalization (DD for DG), and evaluate the unseen domain generalization of models trained on synthetic datasets distilled by state-of-the-art DD methods using the DomainBed benchmark. Additionally, we introduce a new method for this task, which interprets DD through the lens of image style transfer, achieving superior performance in unseen domain generalization compared to baseline approaches.

## 1 Introduction

To mitigate the burdensome costs associated with large-scale datasets such as storing, transmission, and model training, Dataset Distillation (DD) (Wang et al., 2018) is introduced, which aims to reduce the size of a dataset by synthesizing a smaller dataset that approximates the original dataset in terms of the performance of the model trained on it. The DD methods simulate the training process of the model on the original dataset for optimizing the synthetic dataset (Zhao et al., 2021; Cazenavette et al., 2022; Zhao & Bilen, 2023), whose are applied to small-scale datasets such as CIFAR-10/100 (Krizhevsky & Hinton, 2009) and Tiny-ImageNet (Le & Yang, 2015). These methods have been applied to many downstream tasks, such as continual learning (Yang et al., 2023; Gu et al., 2024), privacy preservation (Dong et al., 2022; Chen et al., 2022), federated learning (Liu et al., 2023; Xiong et al., 2023), and medical imaging (Li & Kainz, 2024). And recently, the advent of the decoupling paradigm (Yin et al., 2023; Loo et al., 2024) has paved the way for the distillation of larger-scale datasets, such as ImageNet-1k/21k (Krizhevsky et al., 2012; Ridnik et al., 2021), which highlights the potential of using the distilled datasets as an alternative to large-scale datasets in practice for model training and research, *etc*.

On the one hand, the model in real-world applications requires the robustness to unseen domains, as the input data distribution might differ from the distribution encountered during training. Domain Generalization (DG) is proposed to address this challenge and numerous methods have been studied to improve the generalization to unseen domains. These methods focus on learning domain-invariant features or designing augmentation processes across multiple seen domains, utilizing techniques such as domain adversarial training (Ganin & Lempitsky, 2015; Ganin et al., 2016; Li et al., 2018b; Zhao et al., 2020), domain-invariant representation learning (Li et al., 2018a), and domain augmentation approaches (Zhou et al., 2021b). In addition, various datasets exhibiting domain shifts caused by changes in viewpoint (Fang et al., 2013), or image style (Li et al., 2017; Venkateswara et al., 2017; Peng et al., 2019) have been proposed to evaluate the performance of these methods. The DomainBed benchmark (Gulrajani & Lopez-Paz, 2021) integrates these datasets, offering a comprehensive evaluation for DG methods, and also shows that Empirical Risk Minimization (ERM) (Vapnik, 1998) is a strong baseline compared to these methods.

In order to use the distilled dataset as an alternative to the original large-scale dataset in practice, the robustness of the model trained on the synthesized dataset to unseen domains is also required. However, it remains unclear how the model trained on the distilled dataset is robust to unseen domains compared to that trained on the original dataset and whether existing DG methods remain effective when applied to distilled datasets. To investigate this, we evaluate the unseen domain generalization

| Datset | ERM | DD across domains | | | DD per domain | | | DD for DG | |
|---|---|---|---|---|---|---|---|---|---|
| | | SRe$^2$L | G-VBSM | ratio (%) | SRe$^2$L | G-VBSM | ratio (%) | Ours | ratio (%) |
| VLCS | 75.67 | 51.74 | 37.18 | 0.039 | 73.90 | 73.45 | 0.117 | 73.70 | 0.046 |
| PACS | 80.76 | 53.93 | 43.50 | 0.058 | 67.54 | 76.05 | 0.175 | 76.77 | 0.066 |
| Office-H | 58.94 | 31.84 | 36.22 | 0.347 | 55.98 | 53.46 | 1.042 | 57.45 | 0.352 |
| TerraInc | 42.77 | 23.85 | 20.45 | 0.034 | 27.29 | 19.77 | 0.103 | 40.62 | 0.037 |

Table 1: The unseen domain generalization performance of the model trained on the original and distilled dataset in the DomainBed benchmark. The DD methods, SRe2L, G-VBSM, and Ours, use ResNet-18 (R-18), ResNet-18/50 (R-18/50), and R-18 as the squeeze model, respectively, and use R-50 as the validation model. ratio (%) indicates the condensation rate of the synthetic dataset.

performance of the model trained on the synthetic dataset distilled from a DG benchmark, and compare them to the model trained on the original dataset. Specifically, we apply state-of-the-art DD methods (Yin et al., 2023; Shao et al., 2024; Sun et al., 2024) to the DomainBed benchmark (Gulrajani & Lopez-Paz, 2021) by two straightforward approaches; (1) distilling a dataset using a model trained on all seen domains (DD across domains), and (2) distilling a dataset using models trained on each seen domain separately (DD per domain). The evaluation result in Table 1 shows the degraded performance of the model trained on the synthetic dataset compared to the model trained on the original dataset, and also shows an underlying trade-off between performance and efficiency. Specifically, the degraded performance of the DD across domains approach shows that the model trained on the synthetic dataset distilled by DD methods using a DG model struggles with learning the feature robust to unseen domains. The small performance degradation of the DD per domain approach shows that the synthetic dataset distilled by DD methods using models trained on each domain remains the potential of unseen domain generalization, *i.e.*, the model trained on them can learn a feature robust to unseen domain via DG methods. Still, this approach incurs high costs that linearly increase with the number of domains in terms of model training and storage.

Inspired by the trade-off relationship between unseen domain generalization performance and dataset distillation efficiency, we propose a novel task, Dataset Distillation for Domain Generalization (DD for DG), and introduce a new approach for this task exploiting the advantages of the two straightforward approaches. Formally, the proposed task, DD for DG, aims to provide a small synthetic dataset that approximates the unseen domain generalization performance of the model trained on the original dataset, *i.e.*, efficiently distills the original dataset, which improves the robustness of the model to unseen domains when trained on it. To achieve this goal, we introduce an approach that constructs a single synthetic dataset across domains and learns a domain transfer module that transfers the synthetic dataset into each domain in the original dataset. This approach consists of two processes: a Domain Transfer Learning (DTL) process and a Domain Style Mixing (DSM) process. The DTL process learns to transfer the synthetic images into the style of each domain in the original dataset by interpreting the dataset distillation loss as a style loss in image style transfer (Gatys et al., 2016; Dumoulin et al., 2017). The DSM process augments the domain transfer process to boost unseen domain generalization performance by mixing the learned domain styles.

Our contributions are summarized as follows:

- We address the unseen domain generalization performance of state-of-the-art Dataset Distillation (DD) methods on the DomainBed benchmark, using two straightforward approaches: DD across domains and DD per domain. The evaluation results highlight the vulnerability of synthetic datasets to unseen domains and the trade-off between generalization performance and distillation efficiency.

- We propose a novel task, Dataset Distillation for Domain Generalization (DD for DG), which aims to provide a small synthetic dataset that approximates the robustness to unseen domains in the original dataset. We also introduce a new approach for this task, consisting of two processes: Domain Transfer Learning (DSM) and Domain Style Mixing (DSM).

- The proposed approach outperforms state-of-the-art DD methods in unseen domain generalization, and we conduct ablation studies to further show the effectiveness of the DTL and DSM processes in enhancing performance.

## 2 RELATED WORK

**Dataset Distillation** aims to produce a small synthetic dataset that approximates the original dataset in terms of the performance of a model trained on that dataset. Building on the first work proposed by Wang et al. (2018), which uses a bi-level optimization approach on small-scale datasets such as MNIST and CIFAR-10, the distillation dataset has been scaled up to the down-scaled version of ImageNet-1k (Krizhevsky et al., 2012) using efficient approaches such as matching the loss gradient (Zhao et al., 2021), training trajectory (Cazenavette et al., 2022), and feature distribution (Zhao & Bilen, 2023). Recently, larger-scale datasets beyond full-scale ImageNet-1k have started to distill with attempts to decouple the optimization process (Yin et al., 2023; Shao et al., 2024; Sun et al., 2024; Loo et al., 2024; Sun et al., 2024). We mainly focus on DD methods that distill full-scale ImageNet-1k in this paper.

The full-scale ImageNet-1k is first distilled by SRe$^2$L (Yin et al., 2023), by decoupling the optimization process into the squeeze and recover processes. They propose to use the batch normalization statistics as a regularization loss to optimize the synthetic dataset with cross-entropy loss. G-VBSM (Shao et al., 2024) extends the recover process of SRe$^2$L using diverse squeeze backbone models and convolutional layer feature statistics and proposes the data densification loss which constraint the synthetic dataset has full rank for the diversity of synthetic images. D3S (Loo et al., 2024) reframes DD as a domain shift problem and uses the KL divergence between the feature statistics of the original and synthetic dataset as a regularization loss. RDED (Sun et al., 2024) replaces the optimization process with a patch-wise scoring, top-k selection, and concatenation process using image patches in the original dataset.

**Domain Generalization** has been introduced to mitigate the performance degradation caused by the domain gap between the source and unseen target domain. Extensive research has explored this task, with most approaches based on the premise that domain-invariant features learned from multiple seen domains can also generalize effectively to unseen domains. This goal is typically achieved through domain adversarial training methods (Ganin & Lempitsky, 2015; Ganin et al., 2016; Li et al., 2018b; Zhao et al., 2020; Li et al., 2018a) and domain-invariant-feature learning (Li et al., 2018a). Data augmentation strategies—commonly used as regularization techniques to prevent overfitting—also have been explored to enhance the generalization of models to unseen domains, such as an augmentation method specifically designed to reduce the domain gap (Volpi & Murino, 2019), employing an off-the-shelf style transfer network for data augmentation (Yue et al., 2019; Zhou et al., 2021a), and introducing input/feature levels perturbations using gradients/style representations (Shankar et al., 2018; Zhou et al., 2021b; 2023). As a benchmark for evaluating the domain generalization methods, DomainBed (Gulrajani & Lopez-Paz, 2021) has been widely used and it also argues that Empirical Risk Minimization (ERM) (Vapnik, 1998) is a strong baseline compared to existing DG methods.

**Style Transfer** is a task that transfers the style of one image to another while preserving the content of the former. From the framework proposed by Gatys et al. (2016), the stylized image is obtained by minimizing the style and content losses from a random noise image, where the style/content losses are calculated from the style/content representations of the style/content image and the target image. The ImageNet-1k pre-trained VGG-19 (Simonyan & Zisserman, 2015) is commonly used as a feature extractor for style representation and content representations, and the former is defined as the feature correlation calculated by the Gram matrix and the latter is defined as the feature map itself. The MSE loss on the style/content representations are used for the style/content losses, respectively, and the style loss can be extended to other forms such as batch norm statistic matching and channel-wise mean and variance matching (Dumoulin et al., 2017; Huang & Belongie, 2017).

To alleviate the limitation of requiring separate optimization processes for multiple styles, a framework that transfers the style of a given content image in a single forward pass from a learned style transfer network is proposed Dumoulin et al. (2017); Huang & Belongie (2017). They transfer the given content image to multiple styles using a channel-wise affine transform in the style transfer network, either by learning the style representation within the network from the given style images or by taking the style image as input and extracting the style representation within the network. Also Wang et al. (2021) points out that the ResNet architecture (He et al., 2016) produces unstable style loss due to the residual connection, and propose a technique to a alleviate this issue.

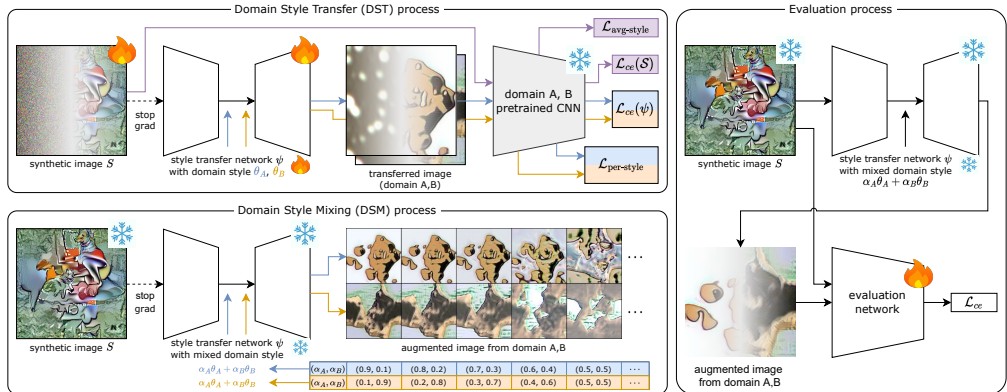

Figure 1: The overall architecture of the proposed Domain Transfer Learning (DTL) and Domain Style Mixing (DSM) processes. $\theta_A$ and $\theta_B$ in the style transfer network $\psi$ denotes the shift and scale parameters for each domain in the conditional instance normalization layer. The relabel process is omitted for simplicity. Best viewed in color.

## 3 DATASET DISTILLATION FOR DOMAIN GENERALIZATION

We propose a new task, Dataset Distillation for Domain Generalization (DD for DG), which aims to distill a small dataset that can train a model robust to unseen domains, and introduce an approach for this task, consisting of two processes: Domain Transfer Learning (DTL) and Domain Style Mixing (DSM), which are explained in the following sections. First, we define the formal definition of the proposed task and the baseline methods in Section 3.1. Then, we introduce the fundamental step of the proposed approach, interpreting the dataset distillation loss as a style loss in image style transfer (Gatys et al., 2016; Dumoulin et al., 2017), in Section 3.2. We then introduce DTL, which learns a domain transfer module to adapt the synthetic images to each domain in Section 3.3. Finally, Section 3.4 explains DSM, which combines learned domain styles to enhance unseen domain generalization performance. The overall architecture of the proposed approach is shown in Figure 1.

### 3.1 PROBLEM DEFINITION AND BASELINES

Given the original dataset $\mathcal{D}$, Dataset Distillation (DD) aims to distill a small synthetic dataset $\mathcal{S}$, subject to $|\mathcal{S}| \ll |\mathcal{D}|$, while maintains the performance of the model $\phi(\cdot)$ trained on the distilled dataset $\mathcal{S}$ in the original dataset $\mathcal{D}$. Formally, the performance of the model $\phi_\mathcal{S}$, which is trained on $\mathcal{S}$, can be expressed as $\mathcal{L}_{\text{ce}}(\mathcal{S}) = \mathbb{E}_{(x,y)\in\mathcal{T}}[-y\log\phi_\mathcal{S}(x)]$, and this leads to the following task formulation:

$$\mathcal{S}^* = \arg\min_\mathcal{S} \mathcal{L}_{\text{ce}}(\mathcal{S}) = \arg\min_\mathcal{S} \mathbb{E}_{(x,y)\in\mathcal{D}}[-y\log\phi_\mathcal{S}(x)], \tag{1}$$

where the model $\phi_\mathcal{S}$ is obtained by $\phi_\mathcal{S}^* = \arg\min_\phi \mathbb{E}_{(x,y)\in\mathcal{S}}[-y\log\phi(x)]$.

As directly optimizing the above formulation requires cumbersome bi-level optimization (Wang et al., 2018), the recent DD methods (Yin et al., 2023; Loo et al., 2024) decouples this process by utilizing the model trained on the original dataset. Specifically, they consider the feature distribution of the original dataset in the trained model as an optimization target for the synthetic dataset, leads to the dataset distillation loss is defined as follows:

$$\mathcal{L}_{\text{DD}}(\mathcal{S}) = \sum_l D\left(p_\mathcal{S}(F^{(l)}), p_\mathcal{D}(F^{(l)})\right), \tag{2}$$

where $F^{(l)}$ denotes the feature maps from the batch normalization or convolutional layer $l$ of the pre-trained network $\phi_\mathcal{D}$ on $\mathcal{D}$, and $p_\mathcal{S}$ and $p_\mathcal{D}$ denote the distributions of the feature map across image $i$ in $\mathcal{S}$ and $\mathcal{D}$, respectively. $D$ denotes the feature distribution matching loss, such as KL divergence or L2 distance, using the summary statistics, such as the channel-wise running mean and variance, which are specified by the DD methods. Note that the cross-entropy loss is also included in the distillation loss, omitted here for simplicity.

The proposed task, Dataset Distillation for Domain Generalization (DD for DG), extends the DD task to distill a synthetic dataset that can train a model robust to unseen domains. Formally, given the original dataset $\mathcal{D} = \{\mathcal{D}_e\}_e$, where $\mathcal{D}_e$ and $e$ for the domain and the domain index, DD for DG aims to distill a small synthetic dataset $\mathcal{S}$ which maintains the performance of the model $\phi_{\mathcal{S}}$ in the unseen domain $\mathcal{T}$. The unseen domain generalization performance of the model $\phi_{\mathcal{S}}$ can be expressed as $\mathcal{L}_{\mathrm{ce}}(\mathcal{S}) = \mathbb{E}_{(x,y)\in\mathcal{T}}[-y\log\phi_{\mathcal{S}}(x)]$, and the task formulation is expressed as follows:

$$\mathcal{S}^* = \arg\min_{\mathcal{S}} \mathcal{L}_{\mathrm{ce}}(\mathcal{S}) = \arg\min_{\mathcal{S}} \mathbb{E}_{(x,y)\in\mathcal{T}}[-y\log\phi_{\mathcal{S}}(x)]. \tag{3}$$

To evaluate the unseen domain generalization performance of $\phi_{\mathcal{S}}$ in $\mathcal{T}$, we apply the recent DD methods as a baseline with two approaches: DD across domains and DD per domain. In the first, a synthetic dataset is distilled using a model $\phi_{\mathcal{D}}$ trained on all seen domains $\mathcal{D}$, formally $\mathcal{S}^* = \arg\min_{\mathcal{S}} \mathcal{L}_{\mathrm{DD}}(\mathcal{S})$. In the second, a synthetic dataset is distilled separately for each domain using models $\phi_{\mathcal{D}_e}$ trained on each domain $\mathcal{D}_e$, formally $\mathcal{S}_e^* = \arg\min_{\mathcal{S}_e} \mathcal{L}_{\mathrm{DD}}(\mathcal{S}_e)$.

## 3.2 Dataset Distillation and Style Transfer

The style of an image can be defined as the channel-wise statistic of the feature maps from a pre-trained network (Gatys et al., 2016; Dumoulin et al., 2017). Formally, the style of image $i$ is defined as the mean and variance of the feature maps from the pre-trained network $\phi$, as follows:

$$\mu_i^{(l)} = \frac{1}{H_l W_l} \sum_{hw} F_{ihw}^{(l)} = \mathbb{E}_{hw}[F_{ihw}^{(l)}], \ (\sigma_i^{(l)})^2 = \frac{1}{H_l W_l} \sum_{hw} (F_{ihw}^{(l)} - \mu_i^{(l)})^2 = \mathrm{Var}_{hw}[F_{ihw}^{(l)}], \tag{4}$$

where $F^{(l)}$ denotes the feature maps from the pre-defined style layer $l$, and $\mu_i^{(l)}$ and $(\sigma_i^{(l)})^2$ are the channel-wise mean and variance of $F_i^{(l)}$ for image $i$, respectively.

The batch norm statistics can also represent the style of a particular domain (Dumoulin et al., 2017), defined as the channel-wise running mean and variance of the images within that domain, as follows:

$$m_{\mathcal{S}}^{(l)} = \mathbb{E}_{i\in\mathcal{S}}[\mu_i^{(l)}], \ (s_{\mathcal{S}}^{(l)})^2 = \mathbb{E}_{i\in\mathcal{S}}[(\sigma_i^{(l)})^2], \quad m_{\mathcal{D}}^{(l)} = \mathbb{E}_{i\in\mathcal{D}}[\mu_i^{(l)}], \ (s_{\mathcal{D}}^{(l)})^2 = \mathbb{E}_{i\in\mathcal{D}}[(\sigma_i^{(l)})^2], \tag{5}$$

where $m_{\mathcal{S}}^{(l)}$ and $(s_{\mathcal{S}}^{(l)})^2$ represent the channel-wise running mean and variance of the feature maps in the batch normalization layer $l$ from the synthetic dataset $\mathcal{S}$, respectively, and same for $m_{\mathcal{D}}^{(l)}$ and $(s_{\mathcal{D}}^{(l)})^2$ from the original dataset $\mathcal{D}$. From this perspective, the dataset distillation loss can be viewed as a form of style loss, matching the style of the synthetic dataset to that of the original dataset. For instance, SRe$^2$L (Yin et al., 2023) employs the L2 distance of the channel-wise running mean and variance in batch normalization layer between the synthetic dataset and the original dataset as the dataset distillation loss, the batch norm style loss between them, where $D\left(p_{\mathcal{S}}(F^{(l)}), p_{\mathcal{D}}(F^{(l)})\right) = \|m_{\mathcal{S}}^{(l)} - m_{\mathcal{D}}^{(l)}\|_2 + \|(s_{\mathcal{S}}^{(l)})^2 - (s_{\mathcal{D}}^{(l)})^2\|_2$. Inversely, the batch norm style loss (Dumoulin et al., 2017) utilizes the MSE loss on the channel-wise mean and standard deviation corresponding to batch normalization layer statistic between the target image $i$ and the style image $j$, which is a kind of DD loss between them, where $D\left(p(F_i^{(l)}), p(F_j^{(l)})\right) = \left(\mu_i^{(l)} - \mu_j^{(l)}\right)^2 + \left(\sigma_i^{(l)} - \sigma_j^{(l)}\right)^2$.

## 3.3 Domain Transfer Learning

Inspired by the close relationship between the style loss and the dataset distillation loss, we propose a method to distill the domain-transferable synthetic dataset with the domain transfer network, dubbed Domain Transfer Learning (DTL) process. This process aims to learn the domain-transferable synthetic images and the domain transfer networks, enabling the transfer from the style of synthetic dataset into the style of each domain in the original dataset.

To learn the domain-transferable synthetic images, we set the target style of the synthetic images as the sum of the MSE loss to each domain style in the original dataset is minimized, where $\arg\min_{(m^{(l)}, s^{(l)})} \sum_e \left(m^{(l)} - m_{\mathcal{D}_e}^{(l)}\right)^2 + \left(s^{(l)} - s_{\mathcal{D}_e}^{(l)}\right)^2$, which leads to the simple average of the domain styles. Formally, the averaged domain style loss for domain-transferable synthetic images

---

**Algorithm 1:** Domain Transfer Leaning

---

**Input** : Style of each domain $(m_{\mathcal{D}_e}, s_{\mathcal{D}_e})$, number of class and image per class $(K, I)$, total
iteration step $T$, iteration step per epoch $N$, and batch size $B$.
**Output:** The synthetic dataset $\mathcal{S}$ and domain transfer network $\psi$.
Compute total epoch $E \leftarrow \lceil T/N \rceil$;
Compute the number of batch per class $M \leftarrow \lceil I/B \rceil$;
Initialize the synthetic dataset $\{\mathcal{S}_{k,m}\}_{k=1,m=1}^{k=K,m=M}$;
Initialize the domain transfer network $\psi$;
**for** $e = 1$ **to** $E$ **do**
 **for** $k = 1$ **to** $K$ **do**
  **for** $m = 1$ **to** $M$ **do**
   **for** $s = 1$ **to** $N$ **do**
    Compute the loss $\mathcal{L}_{\mathrm{DTL}}(\mathcal{S}_{k,m}, \psi)$ from the style of each domain $(m_{\mathcal{D}_e}, s_{\mathcal{D}_e})$;
    Update $\mathcal{S}_{k,m}$ and the domain transfer network $\psi$ by the gradients of the loss;
   **end**
  **end**
 **end**
**end**
Construct synthetic datsaet $\mathcal{S}$ from $\{\mathcal{S}_{k,m}\}_{k,m}$;
return $\mathcal{S}, \psi$;

---

is defined as follows:

$$\mathcal{L}_{\text{avg-style}}(\mathcal{S}) = \sum_l \left( m_{\mathcal{S}}^{(l)} - \mathbb{E}_e[m_{\mathcal{D}_e}^{(l)}] \right)^2 + \left( s_{\mathcal{S}}^{(l)} - \mathbb{E}_e[s_{\mathcal{D}_e}^{(l)}] \right)^2. \tag{6}$$

To transfer the style of synthetic images to the style of domains in the original dataset, we use the style transfer network from the previous work (Dumoulin et al., 2017) as the domain transfer network $\psi$, which learns the style of domains by a single CNN network. The domain transfer network $\psi$ takes the synthetic images $\mathcal{S}$ as inputs and transforms the style of the given domain $e$ in the original dataset, and trained by the domain-specific style loss:

$$\mathcal{L}_{\text{per-style}}(\psi) = \sum_{l,e} \left( m_{\tilde{\mathcal{S}}_e}^{(l)} - m_{\mathcal{D}_e}^{(l)} \right)^2 + \left( s_{\tilde{\mathcal{S}}_e}^{(l)} - s_{\mathcal{D}_e}^{(l)} \right)^2, \tag{7}$$

where the $\tilde{\mathcal{S}}_e = \psi(\mathcal{S}; \theta_e)$ denotes the synthetic images transferred to the style of domain $e$ by the learned style representation $\theta_e$, which is the scale and shift affine parameters in the conditional instance normalization layer of the $\psi$. To prevent the unintentional interference between the domain-specific style loss $\mathcal{L}_{\text{per-style}}$ and the averaged domain style loss $\mathcal{L}_{\text{avg-style}}$ to the synthetic images $\mathcal{S}$, we stop the gradient of synthetic images $\mathcal{S}$ before feeding to the domain transfer network $\psi$.

The style of each layer in the pre-trained network $\phi$ may have different scales, which leads to the unstable learning process by focusing on the specific layer and channel that have large scales, especially in the resnet architecture, which produces peaks caused by residual connection, as pointed out in Wang et al. (2021). To alleviate this unstable training, we normalize the style loss by the standard deviation across channel dimension of the target style, which leads to the final normalized style losses as follows:

$$\mathcal{L}_{\text{avg-style}}^{\text{(normalized)}}(\mathcal{S}) = \sum_l \left( \frac{m_{\tilde{\mathcal{S}}_e}^{(l)} - \mathbb{E}_e[m_{\mathcal{D}_e}^{(l)}]}{\text{std}_c \left( \mathbb{E}[m_{\mathcal{D}_e}^{(l)}] \right)} \right)^2 + \left( \frac{s_{\tilde{\mathcal{S}}_e}^{(l)} - \mathbb{E}_e[s_{\mathcal{D}_e}^{(l)}]}{\text{std}_c \left( \mathbb{E}_e[s_{\mathcal{D}_e}^{(l)}] \right)} \right)^2, \tag{8}$$

$$\mathcal{L}_{\text{per-style}}^{\text{(normalized)}}(\psi) = \sum_{l,e} \left( \frac{m_{\tilde{\mathcal{S}}_e}^{(l)} - m_{\mathcal{D}_e}^{(l)}}{\text{std}_c \left( m_{\mathcal{D}_e}^{(l)} \right)} \right)^2 + \left( \frac{s_{\tilde{\mathcal{S}}_e}^{(l)} - s_{\mathcal{D}_e}^{(l)}}{\text{std}_c \left( s_{\mathcal{D}_e}^{(l)} \right)} \right)^2, \tag{9}$$

where the $\text{std}_c(\cdot)$ denotes the standard deviation operator across channel dimension.

Recalling that the dataset distillation loss includes the cross-entropy loss, the cross-entropy losses for the domain-transferable synthetic images and the style transfer network are defined as follows:

$$\mathcal{L}_{\text{ce}}(\mathcal{S}) = \sum_{(x,y) \in \mathcal{S}} -y \log \phi(x), \tag{10}$$

$$\mathcal{L}_{\text{ce}}(\psi) = \sum_{e} \sum_{(x,y) \in \mathcal{S}} -y \log \phi(\psi(x; \theta_e)). \tag{11}$$

The final training loss for the Domain Transfer Learning (DTL) process is defined as follows:

$$\mathcal{L}_{\text{DTL}}(\mathcal{S}, \psi) = \alpha_{\text{ce}} \mathcal{L}_{\text{ce}}(\mathcal{S}) + \beta_{\text{ce}} \mathcal{L}_{\text{ce}}(\psi) + \alpha_{\text{style}} \mathcal{L}_{\text{avg-style}}^{\text{(normalized)}}(\mathcal{S}) + \beta_{\text{per-style}} \mathcal{L}_{\text{per-style}}^{\text{(normalized)}}(\psi), \tag{12}$$

where the $\alpha_{\text{ce}}$ and $\alpha_{\text{style}}$ denotes the hyperparameters for the cross-entropy loss and the style loss on the synthetic images, respectively, and $\beta_{\text{ce}}$ and $\beta_{\text{per-style}}$ denotes the hyperparameters for the cross-entropy loss and the style loss on the style transfer network, respectively.

**Technical Details.** Due to the memory constraint, it is impractical to train the entire synthetic dataset $\mathcal{S}$ in a single batch, so the training is conducted in pre-defined batch sizes, either by iterating over the classes or instances. The former strategy, class-first iteration, is employed in SRe2L (Yin et al., 2023), where each batch contains a single instance across different classes. In contrast, G-VBSM (Shao et al., 2024) uses the latter strategy, instance-first iteration, where each batch contains instances from the same class, and use the data densification loss, which imposes full rank of the batch, for the diversity of synthetic images.

In our work, we jointly optimize the synthetic dataset $\mathcal{S}$ and the style transfer network $\psi$. The class-wise iteration strategy limits the use of data densification loss, while the instance-wise strategy induces a training bias to the domain transfer network $\psi$, as it favor latter classes in the iteration process. To resolve this, we implement an epoch loop on the class-wise iteration strategy by a fixed number of iteration steps for each loop over the synthetic dataset $\mathcal{S}$. This epoch loop is terminated when each synthetic data is trained by the total iteration steps, as shown in Algorithm 1.

### 3.4 DOMAIN STYLE MIXING

Beyond the domain transfer learning, we propose the Domain Style Mixing (DSM) process, which mixes the style of the learned domain style to encourage the model to robust to unseen domain. This kind of attempt to mix the style of multiple domains to boost the domain generalization performance is also proposed in Zhou et al. (2021b), which mixes the style of the images in the batch by the linear interpolation with the mixing ratio $\lambda$ from the beta distribution. Where the previous work mixes the style representation in the CNN feature extractor, which requires the additional component in the model, the proposed method mixes the learned style representation during the style transfer process, which is computed in the style transfer process. The style mixing process itself is similar to the previous work, which mixes the style of the images in the batch by the beta distribution, $\psi(\mathcal{S}; \lambda\theta_e + (1 - \lambda)\theta_{e'})$, where $e$ and $e'$ denote mixing domain indexes and $\lambda \sim \text{Beta}(0.1, 0.1)$ denotes the mixing ratio sampled from beta distribution.

## 4 EXPERIMENTS

### 4.1 EXPERIMENTAL SETUP

To evaluate the unseen domain generalization performance, we utilize the DG benchmark, DomainBed (Gulrajani & Lopez-Paz, 2021), using VLCS (Fang et al., 2013), PACS (Li et al., 2017), Office-Home (Venkateswara et al., 2017), and Terra Incognition (Beery et al., 2018) as datasets to distill and evaluate. The small-scale datasets, such as C-MNIST (Arjovsky et al., 2020) and R-MNIST (Ghifary et al., 2015), are not considered in this work as the dataset distillation task is mainly focused on the large-scale dataset. As baseline DD methods, we use the recent DD methods, SRe2L (Yin et al., 2023), G-VBSM (Shao et al., 2024), and RDED (Sun et al., 2024), which have succeeded in distilling the ImageNet-1k dataset.

The baselines and the proposed approach use the decoupling technique, consisting of three processes: squeeze, recover, and relabel. The detailed settings of these processes, as applied to the

| Datset | IPC | R-18 | | | | R-50 | | | |
|--------|-----|------|------|------|------|------|------|------|------|
| | | SRe$^2$L | G-VBSM | RDED | Ours | SRe$^2$L | G-VBSM | RDED | Ours |
| VLCS | 10 | 51.98 | 30.51 | 68.65 | 72.27 | 48.10 | 32.68 | 68.80 | 69.65 |
| | 50 | 66.87 | 45.27 | 74.65 | 73.97 | 51.74 | 37.18 | 74.68 | 73.70 |
| PACS | 10 | 36.99 | 28.80 | 59.39 | 64.28 | 31.04 | 27.16 | 60.89 | 67.53 |
| | 50 | 62.18 | 51.20 | 76.51 | 75.20 | 53.93 | 43.50 | 76.31 | 76.77 |
| Office-H | 10 | 31.22 | 24.06 | 51.14 | 55.41 | 26.46 | 23.76 | 52.94 | 56.11 |
| | 50 | 44.87 | 34.62 | 56.74 | 58.14 | 31.84 | 36.22 | 56.77 | 57.45 |
| TerraInc | 10 | 8.38 | 9.50 | 32.55 | 34.54 | 10.23 | 13.31 | 27.70 | 36.72 |
| | 50 | 26.24 | 20.64 | 38.70 | 40.36 | 23.85 | 20.45 | 38.53 | 40.62 |

Table 2: The unseen domain generalization performance from the synthetic dataset distilled by baselines and proposed approach. IPC denotes image-per-class on the synthetic dataset. The validation models are ResNet-18 (R-18) and ResNet-50 (R-50).

DG benchmark, are outlined as follows. In the squeeze process, SRe$^2$L, RDED, and our method use ResNet-18 (R-18) (He et al., 2016) as a squeeze model, while G-VBSM employs both ResNet-18 and ResNet-50. All squeeze models are pre-trained on ImageNet-1k and then trained on the dataset from the DomainBed benchmark following the benchmark's and dataset distillation setup. Specifically, the squeeze model is trained by the ERM algorithm with 5000 training steps and 32 training batch sizes, using the Adam optimizer (Kingma & Ba, 2017) with 5e-5 learning rate, (0.9, 0.999) momentum and 0.0 weight decay, and no learning rate scheduler. The data augmentation is applied with the random resized crop, horizontal flip, color jitter, and random grayscale. The batch normalization layer is trained only in the DD per domain approach; otherwise, it is frozen. The random splits of 8:2 per domain are used for training (testing) on seen (unseen) domains, and 20% of the data is used for model selection by validation. In the recover process, SRe$^2$L, RDED, and G-VBSM each use their own recover process. In our method, the Domain Transfer Learning (DTL) process is applied in this process. In the relabel process, SRe$^2$L, RDED, and G-VBSM each use their own relabel process. Specifically, the data augmentation is applied with the random resized crop, horizontal flip, and cutmix (Yun et al., 2019). In our method, the Domain Style Mixing (DSM) process is also applied in this process. In the validation process, all methods use ResNet-50 as a default validation model, and trained on the synthetic dataset following the DomainBed benchmark's and dataset distillation setup. Specifically, all validation models are pre-trained on ImageNet-1k and then trained by 300 epochs with 32 batch size, using AdamW optimizer (Loshchilov & Hutter, 2019) with 5e-5 learning rate, (0.9, 0.999) momentum and 0.0 weight decay, and cosine learning rate scheduler (Loshchilov & Hutter, 2017). The data augmentation follows the relabel process. The batch normalization layer is trained in the DD per domain approach; otherwise, it is frozen. The model selection is performed by validation on seen domains' 20% splits and testing on unseen domain 80% splits.

## 4.2 QUANTITATIVE AND QUALITATIVE RESULTS

The unseen domain generalization performance of the baselines and proposed approach are shown in Table 2, and the distilled dataset on the PACS dataset by them are visualized in Figure 2. From the quantitative result in Table 2, our approach consistently outperforms the SRe$^2$L and G-VBSM baselines by a large margin. Compared with RDED, our approach beats in a low IPC setting with 52.58% and 57.50% on the R-50 validation model, averaged across four datasets, while at high IPC shows marginal improvements from 61.57% to 62.17%. This is because the RDED method constructs a synthetic dataset by merely concatenating the image patches from the original dataset by their own criteria, as shown in Fig 2c, which would show a similar effect to training the model in the original dataset as the IPC increases.

The difference between the synthetic images distilled by the two approaches on the baselines is also shown in Figure 2, where the DD across domains approach combines all seen domain images into the synthetic ones. In contrast, the DD per domain approach contains each seen domain style into each synthetic one. Compared with the baselines, ours shows that the domain-transferable images and the images transferred to each seen domain, highlighting each domain style.

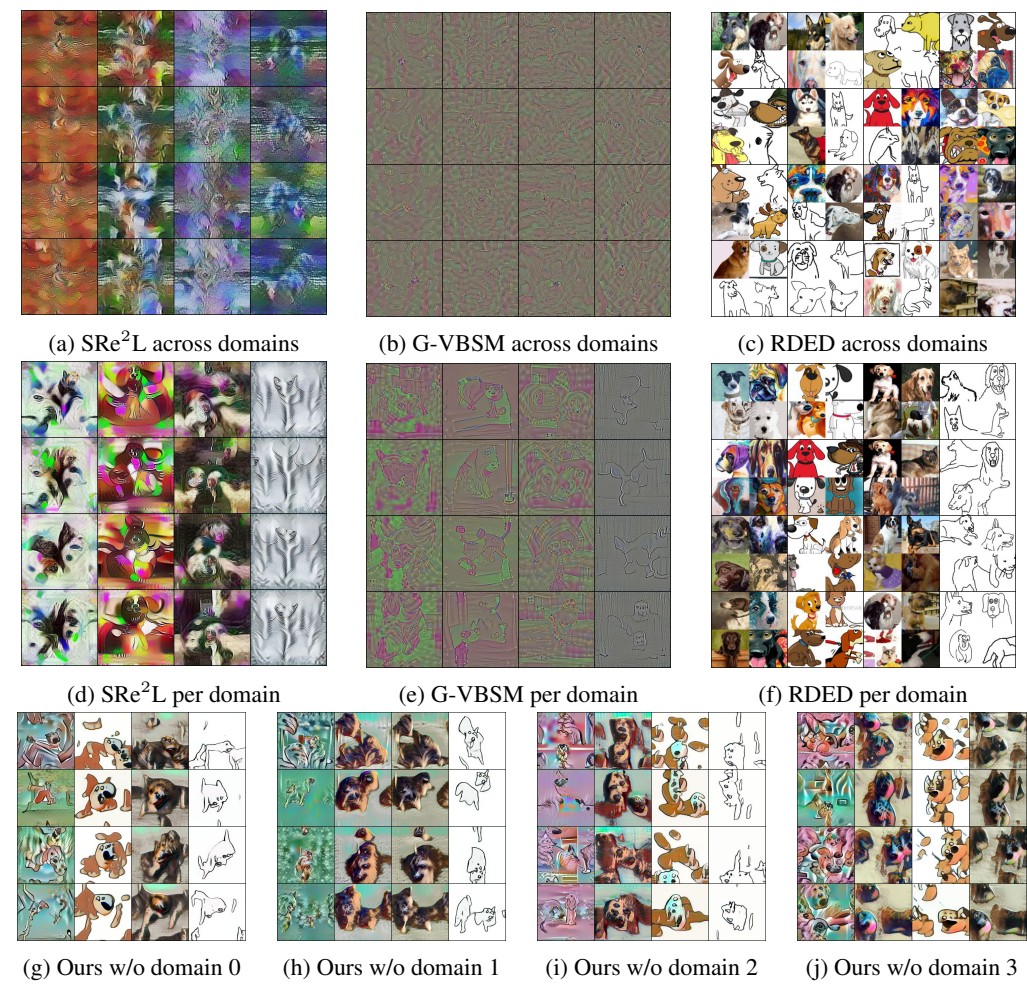

Figure 2: Visualization of distilled dataset by baselines and proposed approach, from PACS art painting, cartoon, photo, and sketch domains. Each row shows the first four synthetic images in the dog class. Each column of the DD-across-domains/DD-per-domain represents the synthetic images for each unseen/seen domain setting, respectively. The 1st and other columns of our approach shows the domain-transferable synthetic image and seen domain transferred images, respectively.

| Dataset | V-16 | R-18 | M-V2 | E-B0 | C-Tiny | D-Tiny | S-Tiny |
|---------|-------|-------|-------|-------|--------|--------|--------|
| VLCS    | 75.54 | 73.97 | 73.15 | 75.34 | 75.34  | 73.36  | 76.60  |
| PACS    | 76.77 | 75.20 | 61.53 | 71.52 | 80.82  | 72.59  | 77.75  |
| Office-H| 55.18 | 58.14 | 54.52 | 56.85 | 60.99  | 53.28  | 59.95  |
| TerraInc| 38.65 | 40.36 | 31.97 | 31.95 | 41.67  | 32.52  | 40.94  |

Table 3: Cross-Architecture Generalization in IPC 50 setting. V, R, M, E, C, D, and S denote VGG, ResNet, MobileNet, EfficientNet, ConvNeXt, DeiT, and Swin, respectively.

## 4.3 Cross-Architecture Generalization

The synthesized dataset is also required to generalize well to the unseen architecture during the synthesizing process. To demonstrate the generalization performance of the synthetic dataset to unseen architecture, the cross-architecture generalization result of our method is shown in Table 3. We conduct the experiment on the IPC 50 setting and use ResNet-18, ResNet-50, MobileNetV2 (Howard et al., 2017), EfficientNet-B0 (Tan & Le, 2020), ConvNeXt-Tiny (Liu et al., 2022), DeiT-Tiny (Touvron et al., 2021), Swin-Tiny (Liu et al., 2021) as validation models. The results show that the distilled dataset by our approach also generalizes well to the unseen architecture.

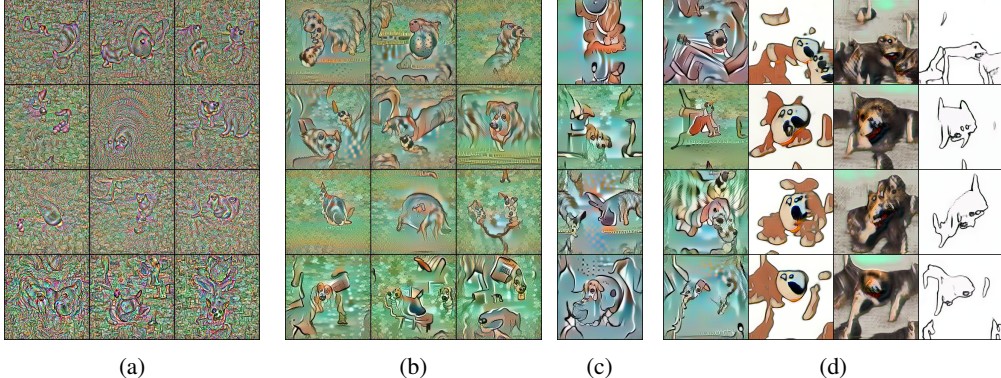

| (a) | (b) | (c) | (d) |

Figure 3: Visualization of the ablation study on the PACS dataset. Each row shows the first 4 synthetic images in the dog class, and each column represents the synthetic images learned by the following style losses; (a,b) the (normalized) domain-specific style loss for Art, Cartoon, and Sketch domains, (c) the averaged domain style loss, and (d) the averaged domain and domain-specific loss with domain transfer network.

| $\mathcal{S}$ | $\psi$ | $\lambda$ | VLCS | PACS | Office-H | TerraInc |
|---|---|---|---|---|---|---|
| $\mathcal{L}_{\text{per-style}}$ | | | 58.12 | 64.63 | 37.82 | 31.60 |
| $\mathcal{L}_{\text{per-style}}^{(normalizd)}$ | | | 72.58 | 74.84 | 55.94 | 28.59 |
| $\mathcal{L}_{\text{avg-style}}^{(normalizd)}$ | | | 71.29 | 71.39 | 56.73 | 28.29 |
| $\mathcal{L}_{\text{avg-style}}^{(normalizd)}$ | $\mathcal{L}_{\text{per-style}}^{(normalized)}$ | | 72.95 | 76.94 | 57.50 | 39.93 |
| $\mathcal{L}_{\text{avg-style}}^{(normalizd)}$ | $\mathcal{L}_{\text{per-style}}^{(normalized)}$ | ✓ | 73.70 | 76.77 | 57.45 | 40.62 |

Table 4: Ablation Study on the training losses in DST and the mixing process of DSM.

## 4.4 ABLATION STUDY

We conduct an ablation study on the losses in the DTL process and the DSM process to investigate the effectiveness of each part, and the result is shown in Table 4 and Figure 3. We use R-18/50 as a squeeze/validation model, respectively. To demonstrate the effectiveness of the losses DTL, we compare the performance of our method in the following setting. The 1st to 3rd rows in Table 4 show the performance of the synthetic dataset trained with the style loss, the normalized style loss, and the normalized average style loss, respectively. The 1st and 2nd rows show that the normalized style loss is more effective on the synthetic dataset distilled per domain. The 2nd and 3rd rows show that the synthetic dataset distilled per domain is more effective than the synthetic dataset distilled by avg-style. The 4th row corresponds to the DTL process, where the domain-transferable synthetic image is transferred to the style in the original dataset to restore the performance of the 2nd row. The 5th row shows the marginal performance boost of the DSM process, demonstrating that the unseen domain generalization performance mainly comes from the domain transfer process.

## 5 CONCLUSION

We investigate the problem of Domain Generalization (DG) on the synthetic dataset distilled by Dataset Distillation (DD), and propose a novel task, Dataset Distillation for Domain Generalization (DD for DG), which aims to provide distilled dataset that can be train a model robust to unseen domain generalization. We evaluate the unseen domain generalization performance of the model trained on the synthetic dataset distilled by current DD methods, and propose a novel approach consisting of two processes, Domain Transfer Learning (DTL) and Domain Style Mixing (DSM), by interpret the current DD method from the perspective of Style Transfer. The experimental results and ablation study show that the proposed approach can improve the unseen domain generalization performance of the model trained on the synthetic dataset distilled by our approach.

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
