# OpenReview forum: "Dataset Distillation for Domain Generalization"
_ICLR.cc/2025/Conference — ICLR 2025 Conference Withdrawn Submission_

### Official Review · Reviewer_oiLy · 2024-11-02

**Soundness:** 2
**Presentation:** 1
**Contribution:** 2
**Rating:** 3
**Confidence:** 3

**Summary:**

- This paper presents a new task, called dataset distillation (DD) for domain generalization (DG). As the name suggests, the goal is to distill multiple source domains via dataset distillation so that the model trained on the distilled dataset exhibits domain generalization properties (good performance on unseen domains). This is a challenge because simply applying dataset distillation to the combined source domains seems to be ineffective.
- The paper proposes domain transfer learning and domain style mixing as the main approach for dataset distillation in DG. Domain transfer learning aims to create the synthetic dataset by matching the style features of different domains, while domain style mixing enhances this by mixing the style features of different domains.
- Experiments are conducted on a subset of the DomainBed benchmark, suing VLCS, PACS, Office-H, and TerraInc.

**Strengths:**

1. The results in Table 1 are show that there is a difficulty in directly applying well-known Dataset Distillation techniques directly to the multi-source DG setting. It clearly shows that naively applying DD across domains does not generate synthetics datasets that can be useful for DG purposes.
2. The authors make an effort to focus on larger scale dataset settings (although I do not agree with what they call "large scale", which will be addressed under weaknesses)
3. The proposed method, especially under low IPC settings, shows some promise.

**Weaknesses:**

Weaknesses are written in no particular order.
1. Figure 1 does not serve much purpose. The only reference of it is in L191, and the caption to support it is short. It's difficult to decipher as a stand-alone figure, but no reference is made in the text (except for "the overall architecture of the proposed approach is shown in Figure 1") so it doesn't help in understand what the proposed method is doing.

2. Using "style" features for domain adaptation or generalization is not a novel concept. It could be argued that using this concept is novel for dataset distillation, but given that the proposed task is a direct combination of DD and DG, I am not convinced that this can be considered as a significant contribution. Minimizing style discrepancies across source datasets to make the model invariant to style variations has been often used for DG. Also, style mixing has been used in many DG methods as well (for example, in "Domain Generalization with MixStyle", as cited by this paper).

3. While the authors claim "small-scale datasets, ..., are not considered in this work as the dataset distillation task is mainly focused on large-scale dataset" (L372), I cannot agree that this paper is conducting experiments on "large-scale" datasets. Let's look at the tested datasets: VLCS, PACS, Office-H, TerraInc. All four datasets use training image size of 224x224 (yes this may be considered as "large image size" compared to MNIST or CIFAR), but they all contain less than 1000 samples. Is this really considered "large-scale"? In contrast, when we talk about dataset distillation scaling to large-scale datasets, we would like dataset distillation to work on ImageNet, which has ~1.2million samples. I'm not really convinced that we need dataset distillation for datasets with <1000 samples. It is also weird that **DomainNet** has been left out from DomainBed, given that DomainNet is the largest dataset in DomainBed.

4. The writing of this paper is quite hard to follow. There are a lot of grammatical errors and typos. Also, some sentences/paragraphs are incoherent or run-on sentences (e.g., L205~207), or just don't make sense (L266: "we set the target style of the synthetic images as the sum of the MSE loss"). Algorithm 1 also seems unnecessary; it gives 4 nested loops followed by two lines of text, which could be simplified to a few sentences in the text itself. Overall, the paper feels very rushed.
  a. Some typos I found: L271, L288, L379, and many more

5. Table 2 should also show results using the original (non-distilled) dataset as a reference to how effective the dataset distillation is. Same goes with Table 3 and Table 1

**Questions:**

- Why was DomainNet omitted from DomainBed? It is the largest dataset in this benchmark.
- Why does the proposed method show more improvement under low IPC settings, compared to higher IPC settings?

---

### Official Review · Reviewer_JNQZ · 2024-11-03

**Soundness:** 2
**Presentation:** 2
**Contribution:** 2
**Rating:** 3
**Confidence:** 4

**Summary:**

This paper introduces novel problem 'Dataset Distillation for Domain Generalisation', which aim to increase the generalisation ability of the model trained on distilled dataset. This new framework is becoming important as the importance of dataset distillation is emerging in recent years. To resolve the issues, the authors provide two main component (1) Domain Transfer Learning (2) Domain Style Mixing, which were inspired by the similarity between dataset distillation process and domain generalisation process. The proposed method increases the generalisation performance on multiple benchmarks.

**Strengths:**

1. The entire problem this paper proposes is quite compelling and yet under explored. Since the dataset distillation technique is gathering attention of ML community for various reasons, it is natural to explore the effects of distilled dataset on final model with various perspectives.

2. The paper is well organised and well written. Especially, the authors well introduce the field of dataset distillation.

3. The authors tried to validate the proposed method across various architectures and dataset.

**Weaknesses:**

1. Lack of validation on large-scale dataset. While the authors presents the experiments with various dataset, most of them are small-size dataset. I believe there are larger dataset for DG tasks, e.g., DomainNet [r1]. While the dataset distillation methods might works not well on large scale dataset, the authors presented that they are now scaled up to ImageNet-1K level. Hence I think the authors should at least validate their method on DomainNet scale, which is quiet smaller than ImageNet but larger than other datasets in the paper.

2. Lack of comparison with existing method. While the authors tries to compare their method with existing dataset distillation method, I can not fine the comparison with existing DG methods. A lot of existing DG methods can be naively incorporated into Dataset Distillation method, e.g., Distill dataset first, then use DG methods when they train network with distilled dataset. I suggest the authors to provide these kind of comparison with existing DG methods.

3. Necessity of domain label. The authors utilise domain labels across their entire process. However, explicitly dividing dataset into separate domains might be impossible in real world environments. Even if it is possible, it requires human expertise, which is quite expensive. I wonder if the proposed method can be used without domain label.

4. Lack of motivation/clarification on DSM. While DSM module is one of main component and maybe boosts the performance, I can not find the motivation related to this paper. Is there any motivation of DSM related to the entire contribution? Also, in the DSM process, do we interpolate the 'weights' of style transfer network? I am asking this because it looks like so, which is quite weird, according to the line362. Also I think the authors should provide comparison results with other kinds of style manipulation methods since there are bunch of style manipulation methods in DG fields, for example StyleMix [r2].


[r1] Xingchao Peng, Qinxun Bai, Xide Xia, Zijun Huang, Kate Saenko, and Bo Wang. Moment matching for multi-source domain adaptation. In ICCV, 2019

[r2] KaiyangZhou,YongxinYang,YuQiao,andTaoXiang.Do- main generalization with mixstyle. In ICLR, 2021

**Questions:**

Please see the weakness section.

---

### Official Review · Reviewer_ewnr · 2024-11-03

**Soundness:** 2
**Presentation:** 3
**Contribution:** 2
**Rating:** 5
**Confidence:** 4

**Summary:**

This paper addresses the challenge of domain generalization (DG) by proposing a novel task called Dataset Distillation for Domain Generalization (DD for DG). The authors highlight the significance of robustness to unseen domains, especially when models are trained on synthetic datasets. They evaluate the unseen domain generalization of models using synthetic datasets distilled through state-of-the-art Dataset Distillation (DD) methods, employing the DomainBed benchmark. A new method is introduced that interprets DD in the context of image style transfer, demonstrating improved performance in generalization compared to existing baseline approaches. The study provides a comprehensive overview of how DD can be leveraged for effective domain generalization in practical applications.

**Strengths:**

1.  The introduction of DD for DG represents a novel perspective in the intersection of dataset distillation and domain generalization, addressing a significant gap in the literature.
2.  The use of the DomainBed benchmark provides a rigorous framework for evaluating model performance across various unseen domains, ensuring that results are robust and generalizable.
3.  The paper discusses the practical implications of using distilled datasets, which can potentially reduce the costs and complexities associated with large-scale data handling in real-world applications.

**Weaknesses:**

1.  While the paper showcases improvements in specific benchmarks, it may not sufficiently explore how these methods perform across a wider range of datasets or real-world scenarios outside of the chosen benchmarks, such as auto-driving scenarios .
2. The efficacy of the proposed method heavily relies on the quality of the distilled datasets. If the initial synthetic datasets are flawed, the results may not hold.
3. The new method, while promising, may introduce additional complexity in practical applications, potentially limiting its adoption in less technical environments.
4.  The approach may struggle with domain shifts that significantly differ from the training conditions. If the unseen domains introduce characteristics not represented in the synthetic datasets, the models may underperform.
5. Many of the figures and images included in the PDF have low resolution, making them difficult to read and interpret. This can hinder the reader's understanding of key concepts and results presented in the paper.

**Questions:**

1. How do the results compare when applying the proposed methods to diverse datasets not included in the DomainBed benchmark?
2. What specific challenges were encountered in the implementation of the new method, and how were they addressed?
3. Could the findings suggest any adjustments to current practices in dataset generation or augmentation to further enhance model robustness?

---

### Official Review · Reviewer_Nrng · 2024-11-04

**Soundness:** 3
**Presentation:** 2
**Contribution:** 3
**Rating:** 5
**Confidence:** 3

**Summary:**

This paper considers the dataset distillation problem in the context of domain generalization, i.e., generate synthetic datasets which enables to train a domain-generalizable model. The authors propose using domain transfer learning and domain style mixing to solve this problem. Experiments verify the effectiveness of proposed method.

**Strengths:**

- The setting this paper considered is new and meaningful.

- The general idea is reasonable.

- Empirical results are good.

**Weaknesses:**

- The paper is not clearly written. Some details are missing or confusing to me.  For example, the definition of $\mathcal{L}_{ce}(S)$ in Eq. (1) or Eq. (3) seems to be inconsistent with that in Eq. (10).

   How do we achieve $\phi(x)$ in Eq. (10)?

   What parameters are updated using Eq. (12)?

   How do you update $S_{k,m}$ in Algorithm1?

   In Figure 3, both (a) and (b) means the normalized domain-specific style loss. What is the difference between (a) and (b)?

   I see in Algorithm 1, $\phi$ is also output. So what will it used for?

- Some typo exist. For example, in Eq. (8), I think the subscript should be $S$ instead of $\tilde{S}$.

- For the visualization, do you have any insights? I cannot relate the performance with the visualizations.

**Questions:**

Please refer to the weakness part. The authors should clarify all the confusing points and provide necessary details to help the readers understand their method.

---

### Note · Authors · 2024-11-14

I have read and agree with the venue's withdrawal policy on behalf of myself and my co-authors.